# Approaches to Measuring the Activity of Major Lipolytic and Lipogenic Enzymes In Vitro and Ex Vivo

**DOI:** 10.3390/ijms231911093

**Published:** 2022-09-21

**Authors:** Marek Wilhelm, Lenka Rossmeislová, Michaela Šiklová

**Affiliations:** 1Department of Pathophysiology, Center for Research on Nutrition, Metabolism and Diabetes, Third Faculty of Medicine, Charles University, 100 00 Prague, Czech Republic; 2Franco-Czech Laboratory for Clinical Research of Obesity, Third Faculty of Medicine, Charles University, 100 00 Prague, Czech Republic

**Keywords:** adipose triglyceride lipase, hormone-sensitive lipase, acetyl-CoA carboxylase, fatty-acid synthase, activity measurement, inhibitor

## Abstract

Since the 1950s, one of the goals of adipose tissue research has been to determine lipolytic and lipogenic activity as the primary metabolic pathways affecting adipocyte health and size and thus representing potential therapeutic targets for the treatment of obesity and associated diseases. Nowadays, there is a relatively large number of methods to measure the activity of these pathways and involved enzymes, but their applicability to different biological samples is variable. Here, we review the characteristics of mean lipogenic and lipolytic enzymes, their inhibitors, and available methodologies for assessing their activity, and comment on the advantages and disadvantages of these methodologies and their applicability in vivo, ex vivo, and in vitro, i.e., in cells, organs and their respective extracts, with the emphasis on adipocytes and adipose tissue.

## 1. Introduction

Triglyceride (TG) metabolism plays an important role in maintaining whole-body energy homeostasis [1]. Regulation of TG amount within cells and particularly in adipocytes requires a carefully regulated balance between lipogenesis and lipolysis, the two main processes in TG metabolism [2,3]. Lipolysis is the catabolic process of TG breakdown into glycerol and fatty acids (FA) and takes place mainly in adipose tissue (AT), liver, and cardiac or skeletal muscle [4,5]. Lipogenesis includes the processes of FA re-esterification and de novo lipogenesis (DNL) [6,7]. Re-esterification of FAs and formation of TGs occur by condensation of FAs and monoglycerides (MGs), diglycerides (DGs), or glycerol phosphate during lipolysis or during food intake [8]. DNL is a metabolic pathway leading to the synthesis of FAs and subsequently TG mostly from glucose. This pathway is essential—especially in the liver—but its important role has also been recently recognized in AT [6,9].

In the last decades, it has been acknowledged that the proper balance between lipogenic and lipolytic processes in AT is necessary for whole body energy homeostasis and that the deregulation of FA metabolism leads to the development of metabolic disorders (as reviewed in [10,11]). Defects in lipolysis in AT observed in obesity may promote lipid overload of non-adipose tissues such as liver, muscle, pancreas, etc. The adverse effects of ectopic lipid overload are summarized under the term “lipotoxicity” [12]. This is associated with the impaired metabolic function of these tissues, insulin resistance, and inflammation [13,14], leading to metabolic diseases such as metabolic syndrome [15], nonalcoholic fatty liver disease [16], and type 2 diabetes [17]. Moreover, excessive TG breakdown also plays a role in cancer cachexia [18]. On the other hand, obesity is associated with increased DNL in the liver, while in AT, DNL is decreased. In the liver, excessive DNL leads to hepatic steatosis linked with lower hepatic insulin sensitivity [19] but, in AT, DNL produces several insulin-sensitizing lipokines [20,21,22] and positively regulates adipocyte membrane fluidity and insulin signaling [10]. The study of the DNL pathway is therefore not only important in basic metabolic research but is also becoming of interest for the targeted treatment of metabolic diseases. These facts explain the urgent need for the use and further development of appropriate methods for studying both lipolytic and lipogenic activity in AT and other organs or cells.

The aim of this review is therefore to provide an overview of the main lipolytic and DNL enzymes, their inhibitors, and methods for determining their activity. We also highlight the advantages and disadvantages of these methods and/or their applicability in metabolic research, especially in adipocytes and AT.

## 2. Lipolysis and Lipolytic Enzymes

In AT, but also in other organs, three major enzymes are involved in lipolysis—adipose triglyceride lipase (ATGL), hormone-sensitive lipase (HSL), and monoacylglyceride lipase (MGL) [5]. TGs, specifically triolein, are very often used as a substrate to measure HSL and ATGL activity despite the fact that HSL hydrolase activity is up to 10-fold higher for DGs compared to TGs [23]. The activity of lipolysis is finely regulated by multiple signals, with catecholamines, insulin, growth hormone, and natriuretic peptides being the main hormonal regulators [10]. The characteristics and regulation factors of key TG hydrolysis enzymes—ATGL and HSL, are described below. Little is known about MGL, which is a 33 kDa membrane enzyme with catalytic triad Ser, Asp, His catalyzing the degradation of DG into glycerol and FA. It is expressed especially in the brain, white AT, and liver [24,25]. It is to be noted that other adipocyte lipases with esterase hydrolytic activity are also involved in the cleavage of TGs [26]; however, they are out of the scope of this review.

### 2.1. Adipose Triglyceride Lipase (EC 3.1.1.3)

ATGL (gene code *PNPLA2*; Patatin-Like Phospholipase Domain Containing 2), also called desnutrin, is a 54 kDa serine hydrolase [23] (catalytic dyad Ser, Asp [27]), discovered in AT by Zimmerman et al. in 2004 [28]. It is expressed not only in AT but also in the liver, testis, skeletal and cardiac muscle, intestine, and β-cells [29,30,31]. It is attached to the lipid droplet by a hydrophobic stretch of amino acids 315–360 [27] and performs the first step in TGs hydrolysis there. ATGL has also been reported to have transacylase and phospholipase activities that appeared to be lower than its TG hydrolase activity [23,29]. ATGL has two phosphorylation sites localized in the C-terminal region of the enzyme (Ser^404^ and Ser^428^). In mouse AT, phosphorylation of these serine residues is carried out by AMP-activated protein kinase (AMPK) and leads to an increase of ATGL activity. In humans, it is not yet fully understood whether ATGL is phosphorylated by AMPK or Protein Kinase A (PKA) [32].

Important regulation of ATGL activity in a cell is provided by a coactivator protein annotated as α/β-fold domain-containing protein 5 (ABHD5), also known as comparative Gene Identification 58 (CGI-58) [33], in coordination with a battery of proteins including the lipid droplet coating proteins from perilipin (PLIN) family. Phosphorylation of PLIN1, which is the major lipid coating protein in adipocytes, is required for the release of ABHD5 from his docking site. ABHD5 then binds to and activates ATGL [34]. PLIN5, which is expressed predominantly in skeletal muscle, binds to both ABHD5 and ATGL itself and activates it independently of PLIN1 [35]. In mice, PLIN2 is known to affect lipolysis only modestly and is not activated by PKA. Overexpression of PLIN2 decreases the access of ATGL onto the lipid droplet and thereby suppresses lipolysis [36]. Data on the role of PLIN3 and 4 in the regulation of ATGL activity are missing. Another coactivator of ATGL is a pigment epithelium-derived factor (PEDF), called by newer nomenclature as serpin F1. The binding of this monomeric 50-kDa protein to ATGL induces TGs hydrolysis [34]. On the other hand, AGTL activity can be inhibited by the protein encoded by G0/G1 switch gene 2 (G0S2) [37] as well as by metabolites, particularly by long-chain acyl-CoA [38].

In addition to biological negative regulators, there are three synthetic molecules that inhibit ATGL activity. One of the first synthetic ATGL inhibitors is Atglistatin, the derivative of urea, which is a selective, competitive inhibitor of ATGL [39]. It is active only in mouse tissues (IC_50_ 0.7 µM) and fails to inhibit more than 10% of ATGL hydrolase activity in human AT [39,40]. Atglistatin affects the patatin-like domain. Sequences 23–101 and 146 amino acids determine the ATGL binding region [41]. Another ATGL inhibitor is Orlistat (tetrahydrolipstatin, ester of L-leucine, and hydroxy-FA with a β-lactone ring). Orlistat is however not a specific inhibitor of ATGL as it inhibits effectively also MGL, HSL, diacylglycerol lipase (a key enzyme in the biosynthesis of the endocannabinoid 2-arachidonoylglycerol [42]), carboxylesterase 1 and even FA synthase (FAS) [43]. Moreover, its major therapeutic potential relates to the inhibition of gastric and pancreatic lipases [44]. Orlistat reduces dietary fat absorption and thus it is currently used as an anti-obesity drug [45,46]. Cay10499 (derivative of methylphenyl ester of N-derived carbamic acid) inhibits ATGL by 95% in humans (IC_50_ 66 nM), but it also inhibits HSL, MGL, diacylglycerol lipase, ABHD6 and, carboxylesterase 1 very efficiently (60–95%) [40]. Recently, a specific inhibitor of human ATGL called NG-497 has been developed. It is a selective, competitiv reversible inhibitor with an IC_50_ of 1.0 µM that binds to human ATGL in a patatin-like domain in the amino acid sequence 60–146. Although this domain is highly conserved between species, even minor differences in amino acid sequence in pigs, primates, and humans vs. other mammals (dog, goat, mouse, rat) apparently substantially affect NG-497 vs Atglistatin efficacy, as shown by Grabner et al. [41]. Thus, species-specific sequence differences (between 60 and 146 amino acids) and the associated structure of the active domain of ATGL are likely to be crucial for the selectivity and efficacy of the inhibitors and, hypothetically, may also serve to regulate ATGL in vivo, which have not yet been revealed. The structure of ATGL inhibitors is shown in Figure 1 and information on them is summarized in Table 1.

### 2.2. Hormone-Sensitive Lipase (EC 3.1.1.79)

HSL (gene code *LIPE*; Lipase E) is—depending on the isoform—84 to 130 kDa [5,23] serine hydrolase, expressed in AT and adrenal glands, with lower expression in cardiac and skeletal muscle and macrophages [23]. HSL consists of three domains. The C-terminal domain contains the active site including the catalytic triad (Ser, Asp, His) [5], while the N-terminal domain interacts with FA binding protein-4 (FABP4). The third domain represents the regulatory stretch [27,111]. HSL performs mainly the second step of TG hydrolysis—it hydrolyses DGs generating MGs and free FA. Besides, HSL hydrolyses also TGs, cholesteryl esters, MGs, and retinyl esters [5,23,27].

The most important activators of HSL are catecholamines, which act through adrenergic receptors and increase the level of cAMP in the cell, followed by activation of PKA [5]. The third domain of human HSL contains serine residues (Ser^552^, Ser^554^, Ser^589^, Ser^649,^ and Ser^650^) that are crucial for the modulation of enzyme activity through their phosphorylation. Phosphorylation of four of these serine residues enhances HSL activity, while phosphorylation of Ser^554^ by AMP-activated kinase inhibits HSL activity [111]. The phosphorylation of HSL and also HSL coactivator—PLIN1—by PKA is necessary for the translocation of HSL from cytosol to the lipid droplet and the initiation of HSL-mediated lipolysis [5,34]. On the other hand, HSL activity may be inhibited upon binding of PLIN2 as shown in cardiomyocytes [62]. As mentioned above, the N-terminal domain of HSL interacts with FABP4, a cytosolic lipid-binding protein expressed in adipocytes, which facilitates FA uptake. The interaction of HSL with FABP4 appears to increase HSL activity after PKA activation [57].

Similar to ATGL, inhibition of HSL has been suggested as a pharmacological approach to reduce free FA levels and improve peripheral insulin resistance. Accordingly, in the last 20 years, various HSL inhibitors, including natural and synthetic products, have been described for the treatment of diabetes and lipid disorders [63]. Moreover, the use of inhibitors and activators of HSL activity is indispensable in basic research for deciphering the HSL’s role in various pathophysiological conditions [112,113,114,115]. The only specific, reversible non-competitive inhibitor of HSL is BAY 59-9435 (BAY), a derivative of 5-(2H)-isoxazolonyl urea. BAY is able to inhibit human, mouse, and/or rat HSL. Nevertheless, while mouse HSL can be inhibited by 90% by BAY, the inhibition of human HSL reaches only 30% [63]. Further, there are several non-specific HSL inhibitors that partially block the activity of HSL, while blocking other lipases as well. Among them is Compound 13f, an ester of 4-hydroxymethyl-piperidine-1-carboxylic acid, which reversibly inhibits human HSL from 17% (5 µM) [40,66] while also acting on other lipases such as carboxylesterase 1, MGL, or ABHD6 [40]. Cay10499 (derivative of methylphenyl ester of N-derived carbamic acid) inhibits mouse and human HSL by 95%, and 67%, respectively, but it also inhibits ATGL, MGL, diacylglycerol lipase, ABHD6, and carboxylesterase 1 very efficiently (60–95%) [40]. Thus, Cay10499 can be successfully used to block all major members of TAG lipolysis, hydrolysis of endocannabinoids, and cholesterol esters [40]. Orlistat also irreversibly blocks, in addition to ATGL, mouse, human, and rat HSL function [40,56,67]. The structure of HSL inhibitors is shown in Figure 1, and information on them is summarized in Table 1.

### 2.3. Assays for Measurement of ATGL and HSL Activity

As mentioned above, ATGL and HSL activity play important role in the regulation of metabolism and pathophysiology of a number of diseases [13,63]. Therefore, the measurement of the activity of these enzymes is widely applied in metabolic research [116,117,118,119].

As the total lipolytic enzyme activity is most conveniently measured at the product level, i.e., glycerol and free FA, the activity of individual lipases can only be analyzed when using purified/isolated enzymes. The only other possibility is to effectively inhibit the activities of all other lipases or to use an enzyme-specific reaction, if one exists. For example, to measure the activity of ATGL, the lipase activity of HSL and other lipases must be inhibited [40,120]. A scheme of inhibitor action on lipolytic enzymes is shown in Figure 2. It is also possible to focus on and analyze one of the esterase activities that none of the other lipases exhibit.

#### 2.3.1. Isolation and Purification of the Lipolytic Enzymes

Isolation of lipases is carried out by disrupting cells or tissue with a Potter-Elehjem homogenizer, freezing, or sonication [120,121,122,123,124,125,126,127,128,129,130,131,132]. This is followed by centrifugation, the conditions of which can vary considerably across sample types, i.e., 1000–110,000× *g*, 1–90 min, 4 °C [120,121,122,123,124,125,126,127,128,129,130,132,133,134,135,136]. After homogenization, recombinant or tissue-derived lipase can be purified by Q-sepharose chromatography, phenyl-sepharose chromatography, QAE-sephadex chromatography [121,125], hydroxyapatite chromatography, or gel filtration chromatography with triacylglycerol-containing acrylamide and agarose (Ultrogel AcA 34) [125].

Another approach to isolating lipolytic enzymes is to isolate lipid droplets that contain various lipases and their co-activators [137]. Isolation of lipid droplets has been successfully performed across many cell types, even on some tissues such as rat and mouse liver, bovine and mouse mammary glands, and others [138]. To homogenize the sample, a homogenizer can be used, or high pressures can be applied using nitrogen bombs. After gentle spinning, the supernatant is removed and ultracentrifuged at 10,000–182,000× *g*, 30–60 min, 4 °C [138,139]. Ultracentrifugation with a density gradient of 5% sucrose can also be used [139]. It should be kept in mind that the higher the speed during centrifugation or its duration, the higher the risk for the destruction of lipid droplets and also removal of proteins from lipid droplets.

#### 2.3.2. Conditions Affecting the Ex Vivo Activity of ATGL and HSL

ATGL and HSL are neutral lipase with a pH optimum of 7.0 [120,121]. Remarkably, when the pH is changed by ±1, the activity of ATGL decreases by almost 50% [120]. The activity of ATGL/HSL can also be inhibited by the detergents used; therefore, it is necessary to dilute the lysate appropriately to avoid detergent interference [120,121,125]. Coactivators can be used to increase enzyme activity. ATGL activity increases 2–20-fold in the presence of coactivator ABHD5 [33,120,123], and is highest when is ABHD5 is emulsified with phosphatidylcholine/phosphatidylinositol (PC/PI) [120].

It should be noted that the isolated/purified enzyme may not have its activators and other regulatory molecules available, so the measured activity is unlikely to reflect the actual in vivo activity. In order to achieve the most physiological situation, it would be advisable to have the activators, metabolites, and ions present in the reaction mixture, preferably in physiological concentrations.

#### 2.3.3. Radioisotope Methods for Measurement of Lipolytic Enzyme Activity Determination

In fact, a radioisotope assay with a radiolabeled [9,10-^3^H]-triolein emulsified with a PC/PI mixture [30,122,123,124,125,126,130,131,132,133,134,140] or dissolved in toluene [120,135] is one of the earliest methods used to measure ATGL or HSL activity. The method is based on the hydrolysis of triolein to free [9,10-^3^H]-oleate and analysis of this product by liquid scintillation counting [30,120,122,123,124,125,126,129,130,131,132,133,134,135,140]. The higher specificity of HSL towards DG may influence the resulting activity. In unpurified lysates, it is necessary to inhibit lipases outside of our interest, as discussed above.

HSL activity can also be measured by substrate specificity to cholesterol esters and DGs (for example, neither MGL nor ATGL have cholesteryl esterase activity [23]). In the first case, the radiolabeled substrate cholesteryl-[1-^14^C]-oleate is hydrolyzed to free cholesterol and [1-^14^C]-oleate [129,132,133,135,136]. In the second case, a radiolabeled [^3^H]-oleoylmonoalkylglycerol, such as [^3^H]-oleoyl-2-O-oleylglycerol, is hydrolyzed to monoalkylglycerol and [^3^H]-oleate [120,121,128]. This substrate has several advantages: firstly, DG is the preferred lipid substrate for HSL, and secondly, monoacylmonoalkylglycerol does not form a substrate for MGL and therefore MGL activity does not interfere in this assay [121].

Generally, in all the mentioned radioisotope methods 0.2–1 mg/mL of total protein is used in the reaction mixture [30,130,132] and the reaction is terminated after 20–60 min of incubation at 25–37 °C [30,120,121,122,123,124,126,127,128,129,130,131,132,133,134,135,136,140] by the addition of a methanol/chloroform/heptane mixture containing 0.1 M potassium carbonate and/or 0.1 M boric acid (pH 10.5) [30,120,123,124,125,126,129,130,131,132,133,134]. After extraction and centrifugation, the radioactivity in the upper phase is determined by liquid scintillation counting [30,120,121,122,123,124,125,126,127,128,129,130,131,132,133,134,135,136,140,141,142].

HSL activity can also be measured using radiolabelled [α-^32^P]-glycerol. In this assay, glycerol released from acylglycerols (TGs, DGs, or MGs) is phosphorylated by glycerol kinase using [^32^P]-ATP. After precipitation of the free [γ-^32^P]-phosphate with ammonium molybdate/triethylamine, the radioactivity of the labeled glycerol in the supernatant is measured by liquid scintillation counting [141,142]. The advantage of this method is the high stability of the measured [α-^32^P]-glycerol [141]; however, isotope ^32^P is a β emitter that poses a potential health risk [143]. Since HSL shows esterase activity towards MGs and DGs, the cleavage of MGs to free glycerol by MGL activity needs to be inhibited to increase the specificity of the reaction. Next, another potential bias could be the varying activity of glycerol kinase, which attaches a phosphate group to glycerol.

Radioisotope assays are applicable to the analysis of ATGL and HSL activity in a wide range of samples including recombinant ATGL [120,130,133] or HSL [120,121,133] (produced in monkey embryonic kidney cells COS-7 [120,130], Escherichia coli BJ5183 cells [133], insect Spodoptera frugiperda cells [121], and rat hepatoma cell line McA-RH7777 [133]) and endogenous ATGL and HSL isolated from rat adipocytes [142], 3T3-L1 adipocytes [140], L6 myoblasts, mouse peripheral leukocytes [131,132], human [122,126,127,128,129,135], mouse [123,124], rat [125] and/or porcine [134] AT, mouse liver [30,133,136,140], skeletal muscle, testes [124] or intestine [30] (summarized in Table 2).

#### 2.3.4. Spectrophotometric and Fluorescence Methods of Lipolytic Enzyme Activity Determination

Other methods for measuring ATGL or HSL activity are based on spectrophotometric [144] or fluorescence [40,144] assays. The spectrophotometric test is performed by the esterase cleavage of p-nitrophenyl esters (acetate, butyrate, or laureate) and the concentration of free p-nitrophenol is measured at 405 nm. In a reaction mixture containing 150 µM p-nitrophenol esters, 5 µg/mL of total protein was used. This method was applied to the mouse white and brown AT [144] (Table 2).

A fluorescence assay uses the ability of the two lipases to cleave pyrene-labeled acylglycerols (Table 2) into free pyrene and acylglycerol, when the fluorescence of the released pyrene is measured. The substrate is dissolved in the PC/PI mixture (3:1, *w*/*w*) and 0.25 mg/mL total protein is added to the reaction, which is run for 1 h at 37 °C. The reaction is then terminated with a solution of chloroform:methanol (2:1, *v*/*v*) and HCl, followed by an extraction process and separation by TLC chromatography. Fluorescent spots are detected using a CCD camera at an excitation wavelength of 365 nm [144]. Another substrate used to measure lipase activity in fluorescence assay is EnzChek (C_58_H_85_BF_2_N_6_O_6_), the commercially developed fluorescent analog of TG. This substrate is successfully cleaved by recombinant human and mouse ATGL or HSL overexpressed in 293T cells (Table 2) and has the advantage of measuring the time dependence of enzyme activity. The reaction mixture contained 2 to 4 mg/mL of total protein. After 30 min preincubation at room temperature, 5 µL of 20 µM EnzChek lipase substrate was added to the reaction mixture, and fluorescence decline was recorded every 30 s for 60–90 min at excitation and emission wavelength at 485 and 510 nm, respectively [40].

HSL activity can be determined also upon the hydrolysis of 1-S-arachidonoylthioglycerol as a substrate. Released 1-thioglycerol spontaneously reacts with ThioGlo-1 to form a fluorescent adduct. The reaction mixture contains 0.11 mg/mL of recombinant HSL and an increase in fluorescence is continuously recorded at excitation and emission wavelength 380 and 510 nm [40].

#### 2.3.5. Advantages and Disadvantages of the ATGL and HSL Activity Measurement

When choosing a particular radioisotope, spectrophotometric and fluorescence method for measuring the activity of lipolytic enzymes, some specifics should be considered. Liquid scintillation radioassay can be used for a variety of tissues or cell samples, whereas spectrophotometric and fluorescence assay has only been described in tissues where the lipases were overexpressed (293T cells or mouse adipose tissue). Considering the amount of total protein needed for the measurement, the methods are quite similar (0.2–1 mg/mL), with exception of the spectrophotometric assay with p-nitrophenol, which uses only 5µg/mL of total protein. The advantage of fluorescence and spectrophotometric assays is the simplicity of both sample processing and the actual measurement, while the certain disadvantage is the lower sensitivity and lower specificity given by using substrates, which are not primary targets of the enzymes and combined specificity of the enzyme for different substrates—contamination by other enzymes. On the other hand, the spectrophotometric and fluorescence methods have the advantage of measuring kinetics parameters in real-time or end-point mode, while radioisotope methods allow only end-point mode.

A number of isolation procedures have been described for the enrichment of ATGL and HSL in the sample prior to activity assays. After the homogenization of biological material, it is recommended to use chromatographic methods with elution techniques that separate proteins on the basis of size. These methods have the advantage that they can preserve the activity of these enzymes; however, a lysate with proteins of similar molecular weight is obtained. A pure enzyme can be obtained by using antibodies against the enzyme, but this may in turn reduce its activity. In general, however, the higher the number of purification steps, the more the activity decreases. Thus, the compromise between the purity of the enzyme and its activity must be considered.

To increase the specificity of the individual enzymes in these methods, specific substrates or inhibitors suppressing the activity of other lipases are used. Of course, both ATGL and HSL prefer TGs/DGs rather than other esters and it must be taken into account that the activity measured with other substrates will always be slightly lower than if measured with preferred substrates.

Indeed, due to the use of unnatural substrates, the activities measured by the spectrophotometric assay do not match the in vivo activities of the measured enzymes. Radioassay requires a number of additional steps such as termination and extraction, which increases the time and cost burden of the method. Moreover, working with radioactive material requires special approvals, and the detection systems are often very expensive. Fluorescence methods are a middle ground between the simplicity of spectrophotometric methods and the variety of applications of biological sample radioassays. However, similarly to in the case of spectrophotometric methods, the use of non-natural substrates probably limits the activity of the enzymes. In fact, the spectrophotometric and fluorescent methods are predominantly used to control the loss of activity of the enzyme during purification steps. Therefore, despite the time and cost involved, radioactive assays appear to be the method of choice when it comes to actually measuring enzyme activity.

## 3. Enzymes of De Novo Lipogenesis

The DNL pathway is of particular interest as it plays a role in non-alcoholic fatty liver disease [173,174], obesity [6,175], insulin resistance [174], or cancer [176]. The process of DNL could be divided into several steps. First, malonyl-CoA is synthesized from acetyl-CoA by acetyl-CoA carboxylase (ACC). The next step is the formation of palmitate by fatty acid synthase (FAS) [177]. Further steps of FA formation from palmitate, including those performed by elongases and desaturases, have been described in detail elsewhere [178,179]. The newly formed FAs are bound to glycerol by mono/diacylglycerol acyltransferase and stored as TGs in fat droplets [180].

### 3.1. Acetyl-CoA Carboxylase (EC 6.4.1.2)

Mammalian ACC (gene code *ACACA/B*; Acetyl-CoA Carboxylase Alpha/Beta) is a homodimeric 265–280-kDa (per subunit, depending on isoform) ligase. ACC catalyzes the rate-limiting step of DNL, the formation of malonyl-CoA, and to do so it uses biotin as a cofactor [9,73]. It contains three major domains—biotin carboxylase, biotin-containing carboxyl carrier protein and carboxyltransferase domain (carboxylation of acetyl-CoA) [78,87]. In mammals, there are two isoenzymes with distinct physiological roles: cytosolic ACC1 participates in DNL, while mitochondrial ACC2 is involved in the negative regulation of mitochondrial oxidation. ACC1 is expressed in lipogenic tissues, including the liver, AT and mammary gland, unlike ACC2, which is expressed to a lesser extent than ACC1 and plays an important role mainly in the heart and skeletal muscle [73].

Specific ACC activity is rapidly activated by insulin and inhibited by catecholamines and glucagon [73,152]. The activity of ACC is also increased by allosteric activators such as citrate [68] or glutamate [70] and/or inhibited by malonyl-CoA, free CoA, and long acyl-CoA esters. Post-translationally ACC is activated by the phosphorylation of several serine residues (Ser^79^, Ser^1200^, Ser^1215^) by AMPK. ACC activity is also modulated by its ability to polymerize [181]. During insulin exposure, the proportion of active ACC polymers increases while catabolic hormones reduce ACC polymerization. Polymerization is also activated by allosteric effectors of ACC [182].

ACC is becoming a potential target in the treatment of a number of diseases mentioned above, namely insulin resistance, dyslipidemia [84], steatosis with non-alcoholic steatohepatitis [183], or cancer [184]. Despite this fact, to date, there is no specific inhibitor for a particular ACC isoform. All available inhibitors act on both isoforms [78,79]. Due to the great variety of chemical structures of ACC inhibitors encompassing more than 40 molecules, only the most widely used ones are commented on in this review.

ACC inhibitors can be subdivided, apart from the nature of the chemical structure, according to the action of the ACC, i.e., the biotin carboxylase or carboxyltransferase domain effect [78,87]. Macrocyclic polyketide antifungal agents (Soraphen A) and thienopyrimidine derivatives (ND-630, ND-646) interact with the ACC biotin carboxylase domain and block its activity [78]. Soraphen A, a natural product of the bacterium Sorangium cellulosum [79], inhibits lipid synthesis at multiple levels (DNL, elongation of FA, synthesis of polyunsaturated FA) in HepG2 [80,81]. Due to ACC inhibition, it also exhibits anti-cancer activity in cancer cells (LNCaP or PC-3M) [83]. Unfortunately, due to its hydrophobic structure, it is not readily soluble in water and its possible use in medicine is therefore limited [78]. Next, biotin carboxylase domain inhibitors, ND-630 (Firsocostat) and its amid derivative ND-646, have higher biological activity compared to Soraphen A and also have better drug-like properties. They were shown to have a hepatoselective effect- they reduce hepatic steatosis in rats and increase insulin sensitivity [84]. The IC_50_ values for inhibition of human recombinant ACC1/2 by ND-630 are 2.1 [78] and 6.1 nM and by ND-646 are 3.5 and 4.1 nM [86], respectively. Interestingly, ND-630 was originally developed as a substrate for an organic anion transport peptide in liver and was later shown to effectively allosterically inhibit the ACC dimerization and activity [78,82,85]. Its analogue, ND-646, is the next generation targeting ACC inhibition [78,87]. Piperidine (CP-640186) and spiropiperidine derivatives (PF-05175157) are good inhibitors of carboxyltransferase domain of ACC. CP-640186 is a reversible allosteric ACC inhibitor and is usable in a variety of tissues such as mouse, human, rat, and monkey [77,82]. The IC_50_ values for inhibition of ACC1 and ACC2 activity in rat liver and skeletal muscle are 53 and 61 nM, respectively [78]. PF-05175157 was synthesized as a hepatoselective inhibitor of ACC (blocking predominantly hepatic DNL) and it is the first ACC inhibitor in human clinical trials. The IC_50_ value determined for human recombinant ACC1/2 was 98 and 45 nM, respectively [89]. There are also a number of inhibitors for which it is not yet known which ACC domain they affect. These agents include furoic acid derivatives (TOFA) and aryl ether derivatives (haloxyfop, tetraloxydim). TOFA is an irreversible, allosteric ACC inhibitor [91], which effectively inhibits FA synthesis by 67% at a concentration of 62 µM by condensation with free CoA to TOFA-CoA [92]. It exhibits cytotoxic effects in human lung cancer, colon carcinoma cells [92], or human breast cancer cells [93]. The IC_50_ value for inhibition of human recombinant ACC1/2 by TOFA is 7–8 µM [94]. The structure of ACC inhibitors is shown in Figure 3, information on them is summarized in Table 1 and a scheme of inhibitor action on lipogenic enzymes is shown in Figure 2.

### 3.2. Fatty-Acid Synthase (EC 2.3.1.85)

FAS (gene code *FASN*; Fatty-Acid Synthase) is a homodimeric transferase of 273 kDa per subunit. Each monomer contains seven domains required for palmitate synthesis: malonyl-/acetyl-transferase, β-ketoacyl synthetase, β-ketoacyl reductase, β-hydroxylacyl dehydratase, enoyl reductase, and thioesterase [185]. FAS is predominantly expressed in the liver, AT, mammary gland, and lung [186] but it can be found in virtually every cell. FAS synthesizes palmitate from acetyl-CoA and malonyl-CoA using NADPH as the energy equivalent. The acyl residue is extended by two carbons donated from malonyl-CoA during the reaction [9,186].

No physiological co-activators or inhibitors regulating FAS activity have yet been identified. Post-translational modifications were detected in human and mouse breast cancer cell lines: phosphorylation of serine residues and acetylation of lysine amino acid residues. While phosphorylation enhances FAS activity [186], acetylation may promote FAS degradation via the ubiquitin-proteasome pathway [169]. Nevertheless, the regulation of FAS activity is still not completely understood.

FAS expression is, similarly to ACC, relatively high in cancer cells because tumor growth puts tremendous pressure on membrane synthesis [187]. Differences in FAS activity in healthy and cancer cells represent a therapeutic window for anticancer therapy [187,188]. Therefore, inhibitors of FAS activity have become an object of interest in recent decades. The first known FAS inhibitor is cerulenin, i.e., epoxy derivative of dodecadienamide [97]. Cerulenin is a non-competitive irreversible inhibitor binding to cysteine residues of the β-ketoacyl synthetase domain [99]. By reducing FA synthesis, it induces selective cytotoxicity in a number of tumor cells [93,95,96]. Unfortunately, due to the very reactive epoxy group, cerulenin is not applicable in clinical medicine. Therefore, cerulenin derivatives lacking the epoxy group have been developed [104]. One of these derivatives is compound C75 (butyrolactone derivative) [101]. C75 inhibits competitively irreversibly β-ketoacyl synthetase, enoyl reductase, and thioesterase domain [102]. Cerulenin and C75 are specific to human and mouse FAS [100,101], C75 even to rat FAS [101]. Mouse FAS is almost completely inhibited by 100 µM cerulenin [100], while 83% inhibition of rat FAS is achieved by 39 µM C75 [101] (Table 1). The effects of cerulenin on human FAS are particularly well known in tumor cell lines such as SKBr-3, MDA-MB-231, or MCF-7 breast cancer cells. The IC_50_ value is in the range of 70–79 µM [98]. Another cerulenin derivative is C93 (structure not published), which also inhibits irreversibly the β-ketoacyl synthetase domain [106,189]. FAS is also inhibited by substances from the β-lactone group, specifically Orlistat, and Ebelactone A or B. As mentioned above, Orlistat predominantly inhibits gastric and pancreatic lipase [44]. However, it inhibits also the thioesterase domain of FAS in cancer cells [43,104,105]. Orlistat (30 µM) can reduce FA synthesis by up to 75% in PC-3 cancer cell lines [43]. Last, but not least, in the 1990s, an antimicrobial drug called triclosan, 5-chloro-2-(2,4-dichlorophenoxy)phenol, has been shown to have a cytotoxic effect against cancer cells and to be a potential FAS inhibitor [107]. Triclosan reversibly inhibits the enoyl-reductase domain of FAS [107,108]. Thus, the effects of inhibitors are mainly studied in cancer cell lines, with cerulenin and C75 being the most commonly studied inhibitors. A number of plant substances such as phenolics (catechnins, flavonoids, tannins, stilbenes) and terpenoids (ursolic, oleanolic acid) have also been shown to inhibit FAS activity. However, their function appears to be multifactorial, affecting FASN gene expression and other pathways as well [106], hence we do not address them further in this review. For further metabolic research, it would be useful to study the effects of the listed inhibitors in metabolically active tissues such as liver and AT, and/or in their cells—i.e., adipocytes, hepatocytes, etc. The structure of FAS inhibitors is shown in Figure 3 and information on them is summarized in Table 1.

### 3.3. Assays for Measurement of ACC and FAS Activity

Due to the clinical aspects of using FAS as a therapeutic target, it is desirable to measure DNL activity. Several approaches have been used, from measuring the activity of individual ACC or FAS enzymes by radiolabeled substrates and spectrophotometric detection of metabolites participating in the reactions to measuring total DNL using the measurement of FAs formed during the reaction.

#### 3.3.1. Isolation and Purification of ACC

Measurement of a specific ACC isozyme activity is reliable only if the isozyme is available in a purified state [146,151]. Determination of ACC1 activity in crude tissue extracts is inappropriate due to interference with side reactions and contamination with mitochondrial isozyme ACC2 and other mitochondrial enzymes [75]. Still, even in tissue preparations without mitochondrial enzymes, the accurate measurement of ACC enzyme activity is complicated because of the accumulation of malonyl-CoA, which may constrain the carboxylation reaction. When determining ACC activity as both isozymes at once, it is advisable to use at least one method separating low molecular weight proteins, otherwise, large amounts of total proteins (>1 mg/mL) must be present in the reaction mixture [147]. 

Purification of ACC has been described in a number of protocols including ammonium sulphate precipitation, calcium phosphate gel fractionation, diethylaminoethyl-cellulose chromatography [68,146], avidin-sepharose chromatography [145,153], gradient centrifugation, hydroxyapatite chromatography [146] and/or polyethylene glycol precipitation [153]. During purification steps it is advisable to use column centrifugation [68,146,148,153,155] and/or dialysis to increase protein concentration in the samples [68,145,146,153]. It is good to note that the isolation procedures are not aimed at isolating ACC specifically but a particular fraction of high molecular weight proteins. In addition to isolation, the presence of citrate and Mg^2+^ ions are essential to increase ACC activity.

#### 3.3.2. Radioisotope Methods of ACC Activity Determination

The methods for measurement ACC activity are using predominantly radiolabeled sodium or potassium bicarbonate Na(K)H^14^CO_3_ and acetyl-CoA as substrates. This reaction produces 1-[^14^C]-malonyl-CoA, which is detected by a scintillation counter [68,145,146,147,148,149,150,151,152,153]. The reaction mixture contains of 0.67–156 µg/mL of isolated ACC [68,153] or 6 mg/mL of total proteins [151]. After incubation for 1 to 10 min at 37 °C [68,146,148,151,153], the reaction is terminated by acid and the denatured proteins and acid salts are separated by precipitation [68,146,148,151,153]. Radioactivity is measured in the dried sample [146,148,151,153] or the dried sample redissolved in water [68]. Protocols without reaction termination also appear [145,147,152]. This assay was used for measurement of ACC activity in a variety of tissue types and cell lines such as lamb [148] or rat [149,150] AT, chicken [146] or rat [149,153] liver, human or rat skeletal muscle [147], rat hepatocytes [151,152] or Fao hepatoma cells [145] (Table 2).

#### 3.3.3. Spectrophotometric Methods of ACC Activity Determination

Two spectrophotometric measurements of ACC activity are based on the conversion of NAD(P)H into NAD(P)^+^ quantifiable as absorbance loss at 340 nm [68,146,153,155]. The first does not measure directly the production of malonyl-CoA but relies on the conversion of NADPH to NADP^+^ occurring during the incorporation of malonyl CoA into the FA chain by FAS [68,155]. The second assay is composed of several consecutive enzymatic sub-reactions. In the first step, malonyl-CoA is formed as in the previous case. ATP used in this reaction is subsequently regenerated by the breakdown of phosphoenolpyruvate into lactate via pyruvate kinase and lactate dehydrogenase. In the last step, NADH is converted to NAD^+^ and the decrease of absorbance of NADH at 340 nm is measured [146]. The reaction mixture contains 0.67–1.33 µg/mL of ACC [153]. The reaction takes 1–10 min incubation at 37 °C [68,146]. In the reaction, pH of 7.5 [153], the coactivator citrate at a concentration of 2–10 mM and Mn^2+^/Mg^2+^ ions are used to ensure optimal ACC function [68,146]. This method was used to measure the activity of ACC isolated from mouse AT [155], chicken [68,146], and rat [153] liver and they are summarized in Table 2.

#### 3.3.4. HPLC Measurement of ACC Activity

Another method of determining ACC activity is to measure the decrease in acetyl-CoA concentration or the increase in malonyl-CoA concentration by reverse-phase HPLC (Discovery C-18 column, 15 cm × 4 mm × 5 µm) with a Waters 996 photodiode array detector [75]. Samples are incubated for 5–20 min at 25 °C. Subsequently, the reaction is stopped by the addition of HClO_4_. After centrifugation at 10,000× *g*, 3 min, reverse-phase HPLC analysis is applied. To our knowledge, this method was successfully applied in mouse 3T3-L1 preadipocytes [75], but its use has not been described for other biological samples (Table 2).

#### 3.3.5. Advantages and Disadvantages of the ACC Activity Measurement

The advantage of the radioassay is its applicability to a wide range of biological materials. Furthermore, there is no need for the presence of other consecutive reactions that require a different enzyme, thus simplifying the reaction considerably. But as written above, radioactive measurements require some skill and a few extra steps that make the method more expensive and time consuming. In addition, this method can be used to determine activity by end-point measurement only.

The major advantage of the HPLC technique is the possibility to measure a decrease in acetyl-CoA concentration or an increase in malonyl-CoA concentration, which allows determining whether a given decrease in acetyl-CoA is related to malonyl-CoA synthesis and vice versa. However, higher sample purity is still necessary for the proper measurement of ACC activity. This can filter out other interfering reactions. However, it is necessary to provide a sufficient amount of substrate throughout the incubation period. The disadvantage is demanding sample preparation and this method provides only end-point data. Furthermore, this technique was used only on mouse 3T3-L1 preadipocytes; as such, its applicability to other cells or tissues needs to be determined.

In general, spectral methods are mostly used not because of their accuracy but because of their simplicity and low cost. Spectrophotometric methods are suitable for kinetic measurements and are used mainly to assess the purity of the enzyme in the analyzed samples during various purification steps. However, the detection of products through secondary enzymatic reactions (as in the case of measuring ACC activity using lactate dehydrogenase) or through the decrease of reducing cofactors (NADPH, NADH) is a certain disadvantage. In the case of insufficiently pure isolation and contamination by enzymes converting the same metabolites, considerable bias may occur when using these methods.

#### 3.3.6. Isolation and Purification of FAS

Isolation and purification steps for FAS from tissue or cell lysates are similar for all methods of analyzing FAS activity. First, the tissue or cells are homogenized by homogenizer and then the lysate is clarified by centrifugation at 1000–105,000× *g*, for 10–60 min [70,148,155,159,161,163,164,166,167,170,171,172]. Purification steps include ammonium sulphate fractionation [70,159,160,161,164,166,173], calcium phosphate gel adsorption [160,161,164,166], anion exchange chromatography [101], gel filtration chromatography [70,159,160,161,164,166] and/or sucrose gradient density centrifugation [161]. It is advisable to apply dialysis between purification steps to concentrate the enzyme and remove salts [70,159,160]. Purification is not necessary for fluorescence assay; however, certain purification steps are described, such as ammonium sulphate fractionation and ion-exchange chromatography [156]. This fraction is then used in the analysis of FAS activities. Nevertheless, the total protein concentration of this purification is significantly lower than other methods. Importantly, FAS is a homodimeric protein and dissociates easily. Therefore, it is necessary to allow time for FAS units to reassemble during all purification procedures, but also during activity measurements. This is not mentioned in many isolation protocols but also activity measurements and should be watched out for.

#### 3.3.7. Radioisotope Methods of FAS Activity Determination

The most commonly used method for measuring FAS activity employs radiolabeled substrates, specifically 1-[^14^C]-acetyl-CoA [159,160,161,166] and/or 2-[^14^C]-malonyl-CoA [159,160,162,163,164,165], whereby the resulting radiolabeled palmitate is detected. The reaction mixture contains 5–10 µg/mL of FAS [161,162] or 50–100 µg/mL of total proteins [161]. After incubation for 1–30 min at 25–38 °C [159,160,161,162,163,164,165,166], termination by acid, base, or alcohol [159,160,161,163,164,165,166] is applied. Lipid extraction by the non-polar solvent is then performed [159,160,161,163,164,165,166]. After drying the extract and redissolving the sample, the radioactivity is measured by liquid scintillation counting [159,160,161,162,163,164,165,166]. Measuring of FAS activity by the radioizotope method was applied to a range of biological samples from cell cultures (BT474 [162], HepG2 [159], LNCaP [165], H35-BT or primary hepatocytes [163]) to tissues (rabbit mammary gland [160], human [164], mouse [163], rat [161] or pigeon [166] liver) (Table 2).

Other isotope methods are using ^13^C-substrates ([^13^C]-acetyl-CoA and/or [^13^C]-malonyl-CoA) and mass spectrometry (MS) for the labelled product detection. Two different MS-detection systems for FAS activity are described [100,158]. The first system is based on a negative ion chemical ionization gas chromatography—mass spectrometer (GC-MS) using the Finnigan DSQ GC-MS system with a ZB-l column to detect 16-[^13^C]-palmitate carboxylate anions. This assay is sensitive enough to detect the activity of purified FAS at a concentration of 10 µg/mL and was applied for recombinant and crude tissue lysate FAS from mouse liver and mammary gland [100] (Table 2). The advantage of this method is the sensitive assessment of FAS activity in the presence of possible contaminants-proteins, enzymes, or drugs. However, for analysis by GC with MS detection, derivatization of FAs with pentafluorobenzyl bromine and/or diisopropylethylamine is required, which changes the chemical structure and thus facilitates the identification of FAs. This is not the case with the second method, which uses LC-MS with LTQ ORBITRAP XL and Orbitrap Fusion Tribrid detectors [158]. The reaction mixture contains 82.5 µg/mL of FAS isolated from cow mammary glands. After incubation, the samples are extracted by hexane. The advantage of LC-MS analysis is the simultaneous quantification of individual ^13^C-labeled free FAs without the need for sample concentration and derivatization [158].

#### 3.3.8. The Spectrophotometric and Fluorescence Methods of FAS Activity Determination

The spectrophotometric assay was commonly used to determine the purity of FAS activity during isolation of the enzyme from tissues, such as rabbit mammary gland [70,160], lamb [148], mouse [155], or human [171,172] AT, human [164] or rat [161,173] liver or from cell lines, such as 3T3-F442A [167], BT474, MCF-7, MDA-MB-231 breast cancer cells [168], human adipocytes [170] or HEK293T, HCT116, and ZR-75-30 [169]. FAS activity is also often analyzed by spectrophotometry in cancer cell lines, namely BT474, MCF-7, and/or MDA-MB-231 cells [168] (Table 2). The principle of the method is based on palmitate biosynthesis, when the reducing equivalent of NADPH is consumed and its decrease is measured spectrophotometrically at 340 nm [160]. The reaction mixture contains 5–125 µg/mL of FAS [161,173] or 50–500 µg/mL of total proteins [155,161,167,169,171,172]. Before initiating the reaction by adding FAS, it is important to preincubate the enzyme at 37 °C to allow dimerization of subunits and maximize enzyme activity [70,160,161,168,173] as mentioned above.

Another method of measuring FAS activity is the fluorescence assay using 7-diethylamino-3-(4′-maleimidylphenyl)-4-methylcoumarin (CPM) thiol probes. The method is based on the release of CoA from malonyl-CoA and acetyl-CoA substrates during the biosynthesis of FAs. Free CoA binds to CPM, and fluorescence of the CPM-CoA adduct is measured at excitation and emission wavelengths of 405 and 530 nm [156,157]. FAS activity was determined by CPM fluorescence in tissue extracts prepared from human lung and lung cancer tissue, [157], rat liver tissue, and human mammary epithelial cell lines SKBr-3 and/or HepG2 [156] (Table 2).

#### 3.3.9. Advantages and Disadvantages of the FAS Activity Measurement

For radioassays, the same conclusions as described for ACC apply, i.e., it is a sensitive, but costly and time-consuming method. Spectrophotometric assay is not suitable for crude enzyme preparations due to possible endogenous oxidation of NADPH catalyzed by contaminating enzymes. This can make the Spectroassay less specific than the radioassay. On the other hand, when the purification and concentration of the enzyme were applied prior to measurement, spectrophotometric assays could achieve similar specificity to radioassays. This should be confirmed in future studies.

In comparison, mass spectrometry-based assays afford direct and specific monitoring of isotope-labeled products; however, the main limitation of this method is the high price for analysis and the necessity of expensive equipment. Among other disadvantages of GC-MS is the necessary derivatization of palmitate by pentafluorobenzyl bromine and/or diisopropylethylamine, while no derivatization is needed for LC-MS. Nevertheless, the applicability of MS analysis in various types of samples needs to be assessed as -opposed to spectroassay or radioassay—its use has been described only in a low number of sample types.

There is also an assay for FAS activity using fluorescence, the CPM assay. The CPM thiol probe is very effective and the whole assay is fast and relatively inexpensive. However, the CPM probe can bind to various thiol groups present in the reaction mixture, including thiol groups of serum albumin, which may represent significant bias and numerous negative and positive controls are necessary (our unpublished data). The assay has been used successfully on a rather small number of biological samples and can provide only end-point data.

FAS enzyme activity has been measured on most different sample types by radioassay; however, this requires special equipment and the ability to work with radioactive material, thus this method is on the decline and is slowly being displaced by mass spectrometry using stable isotopes (^13^C). Despite the cost per sample, GC-MS or LC-MS techniques are probably the method of choice for future measurements of FAS activity. The applicability of these methods to different tissues and cell types should be tested in future studies.

## 4. Summary

Measurement of lipolytic and lipogenic activity is important for understanding the regulatory mechanisms of metabolic processes and also to assess the effectiveness of putative therapeutics targeting these processes in various diseases. Here, we have provided basic information on the most commonly used methods for determining the enzyme activity of four enzymes of lipid metabolism: ATGL, HSL, ACC, and FAS.

Radioassays are the most widely used over other methods for the determination of enzyme activity, not because of their easy applicability, but simply because they were developed in the second half of the last century. This has provided the ability to measure a range of samples and therefore a highly variable range of biological materials from cell lines to different tissues of a variety of organisms can be determined by this method. Currently, efforts are being made to gradually replace radiological tests with other methods, because working with radioactive isotopes is often risky and requires special laboratory conditions. Another long-used approach is the spectrophotometric determination of enzyme activity. As this is a relatively cheap and fast tool, it is often used to determine sample purity during purification steps. A limitation of this method for use in clinical samples and experiments aimed at enzyme activity analysis is the possible inaccuracies due to contamination by other enzymes and metabolites. Similar to spectrophotometric analysis, fluorescence analysis is relatively cheap and fast. Nevertheless, it would be worthwhile to develop new specific fluorescent probes that better reflect the activity of the enzymes in question and to test their applicability in different types of samples. The use of spectrophotometric and fluorescent assays to measure enzyme activity in clinical samples could also be improved by sufficient purification and concentration of enzymes. The main advantage of these methods is that they can analyze enzyme kinetics.

The method on the rise is mass spectrometry, which detects products of enzyme reactions labeled by stable isotopes. The methods powered by mass detection have enjoyed success across the scientific community, including measuring a number of enzyme products. Although the instrumentation is now much more accessible, the cost per the equipment and also per sample analysis is large, therefore only some laboratories can afford this technique. However, if the cost per sample is reduced in the future and the applicability is verified on more samples, this method has great potential for enzyme activity measurement. The only disadvantage of this method is that enzyme activity can only be measured in the end-point mode.

Last but not least, improved purification procedures and the development of more specific inhibitors are needed. Available purification methods are insufficient for all the above-mentioned enzymes. All purification steps are aimed at isolating only a certain fraction of proteins with a range of molecular weights. Without the use of inhibitors of other enzymes, the desired specificity of the reaction cannot be ensured and thus the measurement of enzyme activity is only approximate. It might be useful to develop methodologies using specific antibodies or ligands of given enzymes for their isolation. In addition, no currently used method provides a means to measure enzyme activity in clinical samples, which have a small volume/weight, and a relatively high concentration of total protein is required. Another important limitation of in vitro assays measuring enzyme activity in isolated/purified samples is that they may correspond to quantification of the amount of enzyme rather than a true assessment of ex vivo/in vivo activity. However, as long as identical purification and isolation procedures are maintained in the assay, it is likely that these methodologies can also be used to assess/compare the biological activity of a given pathway or enzyme in different samples, if they are related, for example, to the amount of tissue used for isolation. However, it is good to bear in mind some ‘bias’ in this comparison.

In summary, the possibilities for determining enzyme activity continue to grow, but there is still a need to improve purification steps, develop selective inhibitors for the enzymes in question, and take methodologies to the next level. The driving force for further progress in this field may be the application in clinical practice. Currently, there is already an inhibitor, Orlistat, which is used to treat obesity. Other inhibitors, mostly for FAS and ACC, are being developed for the treatment of cancer. However, their use is still limited. Thus, from the point of view of diagnostic and therapeutic approaches, it is beneficial to study lipogenic and lipolytic pathways to better understand the pathophysiology of metabolic and other diseases, which may help in the development of new therapeutics or personalized medicine. Research on the metabolism and activity of lipolytic and lipogenic enzymes is therefore essential not only for basic research but also for the development of new clinical approaches.

## Figures and Tables

**Figure 1 ijms-23-11093-f001:**
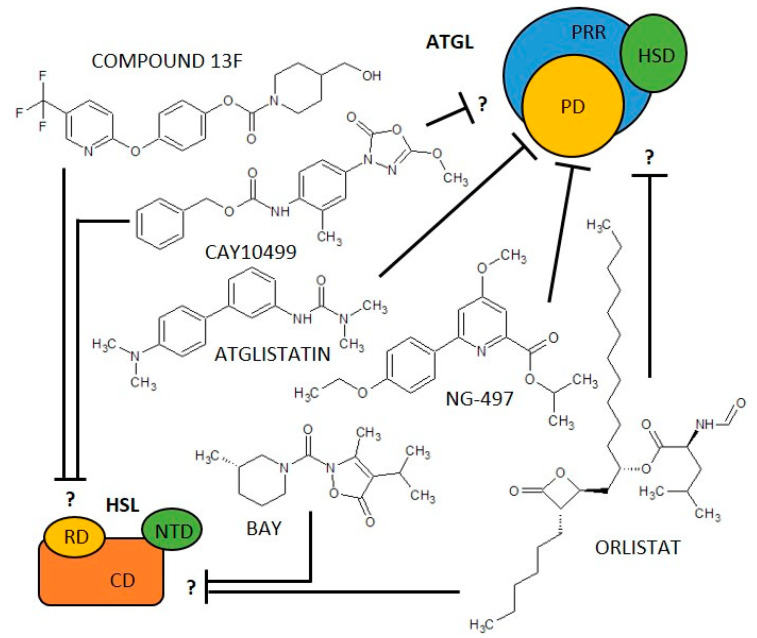
Scheme of adipose triglyceride lipase (ATGL) and hormone-sensitive lipase (HSL) structure/domains and their inhibitors. ATLG domains: PD, Patatin-like Domain; PRR, Patatin Related Domain; HSD, Hydrophobic Stretch Domain. HSL domains: CD, Catalytic Domain; NTD, N-Terminal Domain; RD, Regulatory Domain. ? = it is not described which domain is affected.

**Figure 2 ijms-23-11093-f002:**
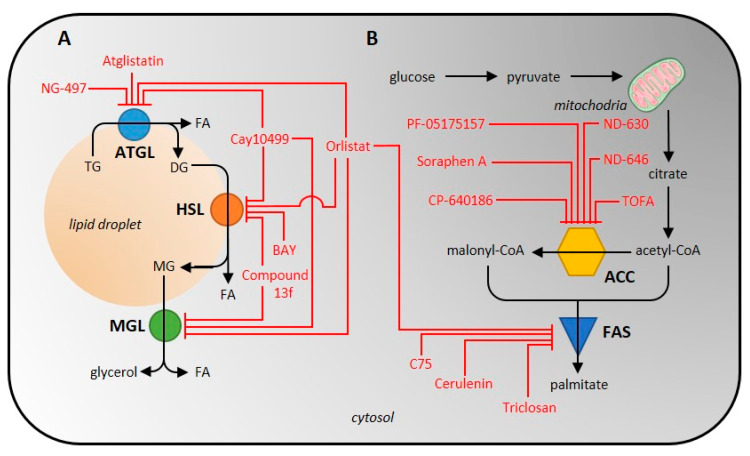
Scheme of inhibitor action on individual enzymes of the lipolytic (**A**) and de novo lipogenic (**B**) pathway. ATGL, adipose tissue triglyceride lipase; HSL, hormone-sensitive lipase; ACC, acetyl-CoA carboxylase; FAS, fatty-acid synthase.

**Figure 3 ijms-23-11093-f003:**
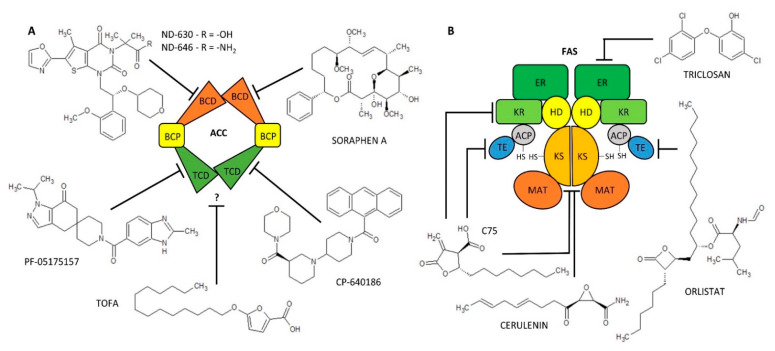
Scheme of acetyl-CoA carboxylase (ACC) (**A**) and fatty-acid synthase (FAS) (**B**) structure/domains and their inhibitors. ACC domains: BCD, Biotin Carboxylase Domain; BCP, Biotin-containing carboxyl Carrier Protein; TCD, Carboxyl Transferase Domain. FAS domains: MAT, Malonyl-/Acetyl-Transferase; ACP, Acyl Carrier Protein; KS, β-Ketoacyl Synthetase; KR, β-Ketoacyl Reductase; HD, β-Hydroxylacyl Dehydratase; ER, Enoyl Reductase; TE, Thioesterase. ? = it is not described which domain is affected.

**Table 1 ijms-23-11093-t001:** An overview of (co)activators and inhibitors of selected lipolytic and lipogenic enzymes.

Enzyme (EC Code)	(Co)Activators	Negative Regulators	(Semi)Synthetic Inhibitors				
			Name(s)	Organism or Cell Line	Enzyme Specificity ^1^	Inhibition Type	IC_50_ [nM]
**ATGL** **(EC 3.1.1.3)**	ABHD5 [33,34,35,47,48] PEDF [34,49,50] PLIN1 [34,35,51] PLIN5 [35,52]	G0S2 [37,48,53] long acyl-CoA [38]	Atglistatin [39,40,41]	dog, goat, marmoset, mouse, rat	yes	competitive, reversible [39]	700 [39]
			CAY10499 [40,54]	human, mouse	no	*unknown*	66 [40]
			NG-497 [41]	human, rhesus monkey	yes	competitive, reversible [41]	1300 [41]
			Orlistat [40,55,56]	human, mouse	no	irreversible [56]	1.2 [40]
**HSL** **(EC 3.1.1.79)**	FABP4 [57,58] PKA [5,59] PLIN1 [5,34]	AMPK [60,61] PLIN2 [62]	BAY 59-9435 [63,64]	3T3-L1, human, mouse, rat	yes	non-competitive, reversible [63]	5 [63]
			CAY10499 [40,65]	human, mouse, rat	no	*unknown*	79.8 [40]
			compound 13f [40,66]	human, mouse	no	reversible [66]	110 [66]
			Orlistat [40,67]	human, mouse, rat	no	irreversible [56]	4230 [67]
**ACC** **(EC 6.4.1.2)**	citrate [68,69] glutamate [70,71,72]	AMPK [73,74] HS-CoA [75] malonyl-CoA [75] long acyl-CoA [75,76]	CP-640186 [77,78,79,80,81,82,83]	human, monkey, mouse, rat	yes	non-competitive, reversible [78]	53–61 [77]
			ND-630, ND-646 [78,82,84,85,86]	human, rat	yes	reversible [84]	1.7–6.1 [84,86,87]
			PF-05175157 [88,89,90]	dog, human, rat	yes	*unknown*	27–33 [88]
			Soraphen A [78,79,80,81,83,87]	HepG2, LNCaP, PC-3M	yes	*unknown*	1–5 [87]
			TOFA [77,78,82,88,89,90,91,92,93,94]	human breast cancer cells, human colon carcinoma cells, human lung cancer cells	no	allosteric, irreversible [91]	51 [94]
**FAS** **(EC 2.3.1.85)**	*unknown*	*unknown*	cerulenin [93,95,96,97,98,99,100]	human, mouse, MCF-7 breast cancer cells, MDA-MB-231, SKBr-3	yes	non-competitive, irreversible [99]	70,000–79,000 [98]
			C75 [96,101,102,103]	HL60 cells, human, mouse, SKBr-3, rat	yes	competitive, irreversible [102]	26,380 [103]
			Orlistat [43,104,105,106]	PC-3, human, mouse	no	irreversible [56]	900 [106]
			Triclosan [107,108,109]	HepG2, MCF-7, SKBr-3 cells	no	reversible [107]	6900–50,000 [107,110]

^1^ Enzyme specificity—yes—the inhibitor is specific to a given enzyme; no—the inhibitor acts on several different enzymes or has several non-specific effects.

**Table 2 ijms-23-11093-t002:** Summarizing the determination of ATGL, HSL, ACC, and FAS activity measurements.

Enzyme	Activity Measurement			Biological Material	Reference
	Detection Method	Substrate	Detecting Substances		
**ATGL**	fluorescence assay	pyrene-labeled acylglycerols	pyrene	mouse AT	[144]
		EnzChek substrate	-	human recombinant protein (HEK 293T cells) mouse recombinant protein (HEK 293T cells)	[40]
	liquid scintillation counting	[9,10-^3^H]-triolein	[^3^H]-oleate	3T3-L1 adipocytes, L6 myoblasts	[140]
				human AT	[122]
				human recombinant protein (COS-7 cells)	[130]
				McA-RH7777	[133]
				mouse gonadal AT	[123]
				mouse liver	[30,133]
				mouse peritoneal macrophages	[132]
				mouse recombinant protein (COS-7 cells)	[120]
				peripheral leukocytes	[131]
	spectrophotometric assay	p-nitrophenyl esters	p-nitrophenol	mouse AT	[144]
**HSL**	fluorescence assay	pyrene-labeled acylglycerols	pyrene	mouse AT	[144]
		1-S-arachidonoylthioglycerol	ThioGlo-1 aduct	human and mouse recombinant protein (HEK 293T cells)	[40]
	liquid scintillation counting	cholesteryl-[1-^14^C]-oleate	[^14^C]-oleate	human AT	[129]
				McA-RH7777	[133]
				mouse liver	[133,136]
				mouse peritoneal macrophages	[132]
		[^3^H]-oleoyl-2-O-oleylglycerol	[^3^H]-oleate	human recombinant protein (Sf9 cells) rat AT	[121] [125]
		[^32^P]-ATP	[α-^32^P]-glycerol	rat adipocytes	[142]
		[9,10-^3^H]-triolein	[^3^H]-oleate	human AT	[126,135]
				mouse AT, skeletal muscle, and testis	[124]
				mouse recombinant protein (COS-7 cell)	[123]
				mouse gonadal AT	[120]
				porcine AT	[134]
	spectrophotometric assay	p-nitrophenyl esters	p-nitrophenol	mouse AT	[144]
**ACC**	HPLC (reverse phase)	acetyl-CoA/ malonyl-CoA	acetyl-CoA/ malonyl-CoA	mouse 3T3-L1 preadipocytes	[75]
	liquid scintillation counting	Na(K)H^14^CO_3_	1-[^14^C]-malonyl-CoA	Fao hepatoma cells	[145]
				chicken liver	[146]
				human skeletal muscle	[147]
				lamb AT	[148]
				rat AT	[149,150]
				rat hepatocytes	[151,152]
				rat liver	[149,153]
				rat skeletal muscle	[154]
	spectrophotometric assay	NADH	NADH	chicken liver	[146]
				mouse AT	[155]
				rat liver	[153]
		NADPH	NADPH	chicken liver	[68]
**FAS**	fluorescence assay	CoA	CPM-CoA adduct	HepG2, human epithelial SKBr-3, rat liver	[156]
				human lung cancer	[157]
	GC-MS/LC-MS	[^13^C]-acetyl-CoA [^13^C]-malonyl-CoA	[^13^C]-palmitate	cow mammary gland	[158]
				mouse liver, mouse mammary gland	[100]
	liquid scintillation counting	1-[^14^C]-acetyl-CoA	1-[^14^C]-acetyl-CoA	HepG2	[159]
				rabbit mammary gland	[160]
				rat liver	[161]
		2-[^14^C]-malonyl-CoA	2-[^14^C]-malonyl-CoA	BT474	[162]
				H35-BT, mouse liver, primary hepatocytes	[163]
				HepG2	[159]
				human liver	[164]
				LNCaP	[165]
				pigeon liver	[166]
				rabbit mammary gland	[160]
	spectrophotometric assay	NADPH	NADPH	3T3-F442A	[167]
				BT474, MCF-7, MDA-MB-231 breast cancer cells	[168]
				HCT116, HEK293T	[169]
				human adipocytes	[170]
				human AT	[171,172]
				human liver	[164]
				lamb AT	[148]
				mouse AT	[155]
				rabbit mammary gland	[70,160]
				rat liver	[101,161]
				ZR-75-30	[169]

## Data Availability

Not applicable.

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
