# Peer review of "Approaches to Measuring the Activity of Major Lipolytic and Lipogenic Enzymes In Vitro and Ex Vivo"

_ijms, 2022, doi:10.3390/ijms231911093_

Round 1

Reviewer 1 Report

The submitted manuscript described in an comprehensive and clear style the available methodologies to evaluate enzyme activity in lipolysis and lipogenesis. There are some minor spelling and grammar mistakes that have to be corrected.

Author Response

We thank to the reviewer for the appreciation of our paper. We went through the entire manuscript and corrected all the spelling and grammatical errors using “Microsoft Word tools”.

Reviewer 2 Report

Comments for ijms-1872709

This manuscript by Wilhelm et al. entitled “Approaches to measuring the activity of major lipolytic and lipogenic enzymes in vitro and ex vivo” well summarizes previously published studies and the current state of understanding on the detecting the activity of major lipolytic and lipogenic enzymes.

One comment: authors could give a perspective discussion on the potential of these approaches for human clinical diagnosis and application.

Author Response

We thank to the reviewer for reading and suggesting further improvement of our manuscript. We have added short discussion on clinical applications to the summary (page 18):

The driving force for further progress in this field may be the application in clinical practice. Currently, there is already an inhibitor, Orlistat, which is used to treat obesity. Other inhibitors, mostly for FAS and ACC, are being developed for the treatment of cancer. However, their use is still limited. Thus, from the point of view of diagnostic and therapeutic approaches, it is beneficial to study lipogenic and lipolytic pathways to better understand the pathophysiology of metabolic and other diseases, which may help in the development of new therapeutics or personalized medicine. Research on the metabolism and activity of lipolytic and lipogenic enzymes is therefore essential not only for basic research but also for the development of new clinical approaches.

Reviewer 3 Report

In this work Whilhelm, M., et. al. review different approaches to measure the activity of lipolytic and lipogenic enzymes in cells and tissues. They discuss the role of each enzyme, methods to purify or inhibit them, different assays that have been used to measures their activity and the limitation of each method. This review is well written and easy to read. Figures and tables are quite helpful. Overall, this work will contribute to a better understanding of these assays and their limitations.

Minor comments

To improve this review, I have some minor comments.

Line 46 - Please provide reference after: “… summarized under the term “lipotoxicity.”

Line 142 - Please provide reference after: “… AMP-activated kinase inhibits HSL activity.”

Line 105: Here the authors discuss different ATGL inhibitor, and they highlight that some of them such as Atglistatin works well for mice tissue but no for human. This is quite true for many inhibitors. I would suggest for the authors to better discuss why inhibitors will have different potency in different species. This would be quite helpful for the readers to think about their experiment.

I would like to suggest that the authors add and discus another ATGL inhibitor that was recent published by Grabner, G.F., et. al. (J. Am. Chem. Soc. 2022, 144, 14, 6237–6250). Here they discuss the limitation of ATGL inhibitors in different species and develop a new small-molecule (NG-497) capable of significantly inhibit primate and human ATGL.  

Author Response

We would like to thank reviewer for his time and invaluable comments.

Citations have been added in lines 46 and 142.

Information on another selective inhibitor, NG-497, was added in the last paragraph of Chapter 2.1 "ATGL" and the differential effect of ATGL inhibitors between organisms was also discussed.

The Table 1, Figure 1 and Figure 2 were also updated.

Reviewer 4 Report

The authors provide an interesting and comprehensive review of the enzymes involved in lipolysis and lipogenesis. They describe the function and structure of the enzymes as well as additional information on their regulation and how to test them experimentally. The text is easy to follow and the figures and tables are synthetic and very informative.

Thanks for this review. I have only some minor comments.

- Some typo errors in the abstract, L129, L151, L158, L215….

-       -   Preferred “Insulin resistance” instead of “resistence”

-       - L161, the sentence must be changed as inhibitors have been already cited the sentences before.

-        -  The exact name/number of BAY should be precised in the main text and table 1.

-       -   §2.3.5. and 3.3.9. As in vitro assays (after protein extraction and/or enzyme purification) mainly require exogenous cofactors or substrates, I think that one of the most important limitations of these "activity" assays is that they correspond more to a quantification of the amount of the enzyme ex vivo/in vivo (determined by its activity) than to a real evaluation of the activity. What is the opinion of the authors on this point? This point could be discussed.

Author Response

We thank the reviewer for his appreciation of our manuscript and suggested improvements.

  1. Some typo errors in the abstract, L129, L151, L158, L215.

All mentioned typo errors were corrected.

  1. Preferred “Insulin resistance” instead of “resistence.

We corrected the spelling.

  1. L161, the sentence must be changed as inhibitors have been already cited the sentences before

Thank you for your comment. This part was indeed not worded fluently. We have changed that sentence, as well as the preceding and following sentences, to make the text clearer.

  1. The exact name/number of BAY should be precised in the main text and table 1.

The BAY number was added to the text and table 1.

  1. §2.3.5. and 3.3.9. As in vitro assays (after protein extraction and/or enzyme purification) mainly require exogenous cofactors or substrates, I think that one of the most important limitations of these "activity" assays is that they correspond more to a quantification of the amount of the enzyme ex vivo/in vivo (determined by its activity) than to a real evaluation of the activity. What is the opinion of the authors on this point? This point could be discussed.

We agree with the reviewer that this limitation is important for in vitro assays.  Therefore we expanded a part of the paragraph in the summary as follows:

Another important limitation of in vitro assays measuring enzyme activity in isolated/purified samples is that they may correspond to quantification of the amount of enzyme rather than a true assessment of ex vivo/in vivo activity. However, as long as identical purification and isolation procedures are maintained in the assay, it is likely that these methodologies can also be used to assess/compare the biological activity of a given pathway or enzyme in different samples, if they are related, for example, to the amount of tissue used for isolation. However, it is good to bear in mind some 'bias' in this comparison.

Round 2

Reviewer 2 Report

Accept

Author Response

We thank to reviwer for the acceptation.